# Natural Enemies of Fall Armyworm *Spodoptera frugiperda* (Lepidoptera: Noctuidae) in Different Agro-Ecologies

**DOI:** 10.3390/insects12060509

**Published:** 2021-05-31

**Authors:** Albert Fomumbod Abang, Samuel Nanga Nanga, Apollin Fotso Kuate, Christiant Kouebou, Christopher Suh, Cargele Masso, May-Guri Saethre, Komi Kouma Mokpokpo Fiaboe

**Affiliations:** 1IPM Unit, International Institute of Tropical Agriculture (IITA), P.O. Box 2008 Messa, Yaoundé, Cameroon; a.abang@cgiar.org (A.F.A.); s.nanga@cgiar.org (S.N.N.); c.masso@cgiar.org (C.M.); k.fiaboe@cgiar.org (K.K.M.F.); 2Agricultural Investment and Market Development Project (PIDMA), MINADER, Yaoundé, Cameroon; kchristiant@yahoo.fr; 3Institute of Agricultural Research for Development (IRAD), P.O. Box 2123, Yaoundé, Cameroon; suhchristopher@yahoo.com; 4International Institute of Tropical Agriculture (IITA), R4D Directorate, PMB 5320, Oyo Road, Ibadan 200001, Oyo State, Nigeria; may-guri.saethre@norad.no; 5Norwegian Agency for Development Cooperation, Bygdøy allé 2, 0257 Oslo, Norway

**Keywords:** indigenous parasitoids, invasive pests, emergency actions, conservative and augmentative biological control

## Abstract

**Simple Summary:**

Fall armyworm (FAW), *Spodoptera frugiperda*, was reported in Cameroon for the first time in 2017 and by the end of 2018; the pest was found all over the country. Cameroon is among the first countries in Africa where the southern armyworm (SAW), *Spodoptera eridania*, another economical important armyworm was reported. The African governments adopted emergency actions around chemical insecticides despite the range of economic and health risks associated with chemical control. This work aims at identifying parasitoids (natural enemies) of armyworms and test their acceptability, suitability, and host range on *Spodoptera* spp., that can play a significant role in the sustainable management of these spodopterans. Field surveys conducted lead to the identification of two egg and four larval parasitoids. The fall armyworm was the predominant spodopteran. Laboratory studies were conducted to assess the performance of parasitoids associated with both pests in Cameroon. *T. remus* showed significantly higher parasitism on FAW than SAW, with significantly shorter development time on FAW, while inducing significant non-reproductive mortality on FAW. Laboratory performance of larval parasitoid was not compared between the two spodopterans identified but the developmental parameters showed that *C. icipe* has a shorter development time compared to other larval parasitoids.

**Abstract:**

Fall armyworm (FAW) *Spodoptera frugiperda* (J.E. Smith) and southern armyworm (SAW) *Spodoptera eridania* (Stoll) have become major threats to crops in Africa since 2016. African governments adopted emergency actions around chemical insecticides, with limited efforts to assess the richness or roles of indigenous natural enemies. Field surveys and laboratory studies were conducted to identify and assess the performance of parasitoids associated with spodopterans in Cameroon. FAW was the most abundant spodopteran pest. *Telenomus remus* (Nixon), *Trichogramma chilonis* (Ishi), *Charops* sp. (Szépligeti), *Coccygidium luteum* (Cameron), *Cotesia icipe* (Fernandez & Fiaboe), and *Cotesia sesamiae* (Cameron) are the first records in the country on spodopterans. *Telenomus remus, T. chilonis*, *C. icipe*, and *Charops* sp. were obtained from both FAW and SAW; *C. luteum* and *C. sesamiae* from FAW. The distribution of spodopterans, their endoparasitoids, and parasitism rates varied with host, season and location. In the laboratory, *T. remus* showed significantly higher parasitism on FAW than SAW, and significant differences in the development parameters between the two host eggs, with shorter development time on FAW. It induced significant non-reproductive mortality on FAW but not on SAW. Developmental parameters showed that *C. icipe* has a shorter development time compared to other larval parasitoids. Implications for conservative and augmentative biocontrol are discussed.

## 1. Introduction

*Spodoptera* Guenée (Lepidoptera: Noctuidae) is a genus that comprises 31 species [1,2]. Species from this genus are commonly known as armyworms among which eight have been reported in Africa since the 1970s [3]. This includes *Spodoptera exempta* (Walker), *S. exigua* (Hübner)*, S. triturata* (Walker)*, S. mauritia* (Boisduval)*, S. cilium* (Guenée)*, S. apertura* (Walker)*, S. littoralis* (Boisduval), and *S. malagasy* (Viette) [4]. *Spodoptera litura* (Fabricius) was reported later from Africa in Ghana on okra [5]. Fall armyworm (FAW) *S. frugiperda* (J.E. Smith) and southern armyworm (SAW) *S. eridania* (Stoll), have recently invaded Africa and causing havoc since 2016 [6,7]. Among these armyworms, only *S. frugiperda* is reported to feed on both monocots and dicots [8], with a host range of about 353 recorded plant species in 76 families [9]. FAW has become the major threat to maize production in Sub-Saharan Africa (SSA) since its first report in Africa. Significant damages to maize in farmers’ fields have been recorded with yield losses estimated around 8.3–20.5 mil tons, equivalent to some US$2.5–6.2 billion from 12 countries in Africa [10]. FAW was reported in Cameroon for the first time in 2017 [11]. The distribution, population genetics, and damage of FAW assessed through a country-wide survey in 2017 revealed its presence in all regions, the existence of two haplotypes in four of its five agro-ecological zones, and damage levels spread equally across these locations [12]. Less is known and reported on the fact that at the same time FAW was being detected in various African countries, the other closely related species, SAW, also of high economic importance, was reported in the continent [6]. SAW distribution, population genetics, and damage in the country and the continent, in general, has not been assessed, although it occurs throughout the Americas, and has been reported on 202 host plant species from 58 families [13]. 

To tackle FAW and in an initial belief of its eradication on the continent, governments and regional bodies across Africa quickly adopted emergency actions predominantly built around the use of synthetic insecticides, with no consideration for potential associations from indigenous natural enemies existing in the system. But their efforts were futile because little success has been registered in the reduction of infestation and damage on maize, yet leading to higher cost of production and development of resistance to pesticides [10,12,14,15]. Four years following the first report of the pest on the continent, it has now been noted that the pest established populations in all the countries where it was reported, denoting that eradication programs and emergencies actions should be replaced by sustainable interventions since the pest is here to stay [16]. While spraying chemical pesticides indiscriminately, limited to no efforts were made in most African countries to assess the beneficial complex, particularly parasitoids, that could adapt and re-associate with the invasive pests on the continent. Yet it is common knowledge that other *Spodoptera* species have been kept under control throughout the continent. Furthermore, considering the polyphagous nature of the pests, we hypothesize that interventions limited to farm level are likely not enough to ensure the sustainable management of the pest. Natural enemies, parasitoids, entomopathogenic fungi, bacteria, viruses, nematodes, as well as predators constitute an immense resource that could offer additional value outside farmed areas by following the pests even in the wild. They are central to the development of integrated pest management (IPM) systems [17]. Indiscriminate use of chemical pesticides could lead to the decimation of the beneficial and lead to an outbreak of pests that were kept under control in the systems over decades. However, to protect these beneficials non-target effect of most pesticides are required but a pre-requisite for this is to know the potential indigenous natural enemies in the system, as well as assess their potential ecosystem services provided against existing potential spodopteran pests, but most importantly their potential association with the newly invasive *S*. *frugiperda* in Africa.

Among the natural enemies, parasitoids have been reported as the most common natural enemies used with biological control of spodopterans being well documented for the Americas, especially in Central America [18] and in the southern part where 86 of the 150 species parasitizing FAW were reported [19,20,21]. Among these parasitoids, *Telenomus remus* Nixon (Hymenoptera: Platygastridae) and *Trichogramma* spp. (Hymenoptera: Trichogrammatidae) are the main egg parasitoids of FAW in North and South America, where they are already used in augmentative biological control [22,23,24]. Around the America, larval parasitoids such as the Braconidae: *Meteorus autographae* Muesebeck, *Meteorus laphygmae* Viereck, *Chelonus texanus* Cresson, *Cotesia marginiventris* (Cresson, 1865), *Aleiodes laphygmae* have been reported on *S. eridania*, while *Aleiodes vaughani* (Muesebeck) share both *S. cosmioides* (Walker) (Lepidoptera: Noctuidae) and *S. eridania* [25]. The Ichneumonidae: *Campoletis flavicincta* (Ashmead) and *Campoletis sonorensis* (Cameron), *Ophion* spp, and *Eiphosoma dentator* (Fabricius) were reported on *S. eridania* in Central and South America and the Caribbean [26,27]. The most recent effort to identify parasitoids of spodopterans is the report by Freitas [27] about the first record of *Cotesia scotti* (Valerio and Whitfield) (Hymenoptera: Braconidae) as a parasitoid of the black armyworm *S. cosmioides* and the Southern armyworm *S. eridania* in Brazil. Similar efforts on the identification of parasitoids of lepidopterans are needed worldwide where the armyworm invaded.

Little is known about the natural enemies of fall armyworm in Africa since it is a recent pest. Reports about parasitism of other spodopterans including the SAW are scarce with most of them being highly polyphagous or associated with FAW also. *T. remus*, an egg parasitoid initially considered for introduction into Africa, has been recently reported to be present in Benin, Cote d’Ivoire, Kenya, Niger, and South Africa [15]. This species is among the 150 parasitoids recorded in the Americas but was introduced from Malaysia to parts of Africa, precisely Cape Verde as well as to America, Asia, and Europe against *S. exempta*, *S. exigua,* and *S. littoralis* [15]. In Ethiopia, three larval parasitoids namely *Cotesia icipe* Fernandez-Triana & Fiaboe (Hymenoptera: Braconidae), *Palexorista zonata* Curran (Diptera: Tachinidae), *Coccygidium luteum* Brullé (Hymenoptera: Braconidae) have been identified [28]. In Kenya, four larval endoparasitoids *Charops ater* Szépligeti (Hymenoptera: Ichneumonidae), *C. icipe, P. zonata*, *C. luteum* and two egg parasitoids *Chelonus curvimaculatus* Cameron (Hymenoptera: Braconidae) and *Trichogramma* sp. (Hymenoptera: Trichogrammatidae) were reported from FAW [28]. In Tanzania, two species *C. ater* and *C. luteum* were reported [28]. In maize and sorghum fields in Niger three egg parasitoids, *Trichogrammatoidea* sp. (Hymenoptera: Trichogrammatidae), *Trichogramma* sp., and *Telenomus* sp.; one egg-larval parasitoid *Chelonus* sp.; four larval parasitoids, *Cotesia* sp., *Charops* sp., and unidentified ichneumonid and tachinid fly are reported on FAW [29]. Although there are no reports of isolated entomopathogens, the efficacy of *Metarhizium anisopliae* and *Beauveria bassiana,* have been shown against eggs and second-instar larvae of FAW in the laboratory [30]. In Cameroon, studies conducted to assess FAW presence, geographic distribution, and levels of infestation and damage, noted the presence of potential natural enemies like spiders, predatory wasps, earwigs, larval, and eggs endoparasitoids [12]. However, these were only preliminary results with their identification limited to family level with confirmation needed. 

This study, led by the International Institute of Tropical Agriculture (IITA), is an effort to identify the diversity and roles of indigenous parasitoids species associated with armyworms on various host plant species and under different agro-ecological zones in Cameroon. The specific objectives are to: (i) Identify the natural enemies associated with armyworms and their parasitism rates; (ii) determine the geographic distribution and host range of indigenous natural enemies; and (iii) assess the biological performance of the parasitoids on their respective spodopteran hosts. This information is critical in developing and achieving efficient and sustainable management of the invasive armyworm pests by integrating conservative and augmentative biological control into an IPM package that not only controls the pest in cultivated areas but also in the wild.

## 2. Materials and Methods

### 2.1. Study Area Description

Seven rounds of field surveys were conducted in Cameroon, three in 2017 as the exploratory phase, three in 2019, and one in 2020 to assess parasitism rates. The first phase was conducted to establish the diversity and distribution of parasitoids and took place between February to March, May to June, and October to November 2017 representing peak maize production periods. Surveyed fields fall within four of the five agro-ecological zones namely the Guinea savannah (Adamawa region) as Zone II, the Western Highlands with savannah vegetation and mono-modal rainfall (West and North-West regions) as Zone III, the humid forest with mono-modal rainfall (Littoral and South-West regions) as Zone IV and warm and humid forest with bi-modal rainfall (Center, South, and East regions) as Zone V (Figure 1). The Soudano-sahelian Zone with erratic rainfall (North and Far North regions) as Zone I was excluded because of security reasons. During the exploratory survey, 420 fields were surveyed in 261 villages grouped in 61 locations (Table 1). After the exploratory phase, four surveys were conducted for the second phase with three in 2019 from May to June, July to August, September to October 2019, and one from January to April 2020. These second and third phases were to determine the rates of field parasitism of armyworm immature stages by various parasitoid species and targeted locations in a radius of 200 km from Yaoundé to cover two agro-ecological zones; the humid forest with mono-modal rainfall (Littoral and South-West regions) which is Zone IV and the humid forest with bi-modal rainfall (Center, South and East regions) which is Zone V (Figure 1). The locations involved were Pouma and Edea in Zone IV and Monatele, Bafia, Ntui, Yaoundé and Mbalmayo in Zone V. However, Bafia and Ntui with savannah vegetation type and almost mono-modal rainfall pattern represent a transition Zone between Zone III and Zone V. However, the rates of parasitism from exploratory phase have been included in this study. During the study to determine field parasitism, 176 fields were surveyed in 72 villages across the six locations or districts (Table 1). For the three years, a minimum of three districts were surveyed per round in each agro-ecological zone visited. In each district at least three fields, one per village were sampled. During the dry season, maize is cultivated mainly in areas with irrigation systems or in valleys and riverbanks with sufficient soil moisture.

### 2.2. Identification of Natural Enemies of the Armyworms

The surveys were carried out on farmers’ crop fields with their permission. At least two districts were visited per region and three per agro-ecological Zone and at least three villages visited per district or location. The number of fields sampled in each location was based on the availability at the time of sampling, presence of armyworm symptoms, and only unsprayed fields were considered. For all surveys, to avoid sampling fields that were close to each other, one field was sampled per village unless constrained by the unavailability of fields then more than one field could be sampled per village. In each field, geographic coordinates (using handheld Garmin GPS) were collected. Before sampling in any field, the owner’s consent was sought and obtained, and the farmer was requested to provide information on the planting date and whether the field was sprayed with pesticides or not. Provided information was treated anonymously. 

Maize or other known host plants around maize fields were inspected, and all larvae and egg batches found were collected from the whole field with field sizes ranging from 50 m^2^ to 0.25 ha. Where fields were bigger, a subplot of 0.25 ha was selected and sampled for larvae collection. Each sampled field was divided into four equal plots and in each plot, scouting was done by inspecting 20 plants, moving along a W-shape design. The middle of the field was also sampled, making 100 plants surveyed per field. Distance between the two consecutive plants was a function of field size and shape but was representative of the plot area [31]. The number of larvae (from neonates to instar 4) and egg batches collected was recorded. Larvae were placed in ventilated plastic transparent 1 L jars (about 15 cm high and 10 cm diameter) with leaves of respective host plants as diet while egg batches were placed individually in vials (7 cm height and 2.5 cm diameter) aerated with plastic lids, for incubation, labeled and taken to the insectary at IITA station in Yaoundé (03°51.791′ N; 011°27.706′ E, 747 masl). Parasitoid cocoons found in the field were collected individually in the same type of vials used for egg batches. The same types of vials were used for the collection of already parasitized egg batches. Parasitized egg batches had eggs that were completely dark (not black-head-stage). All specimens were maintained under laboratory conditions with a photoperiod of 12 L:12D at room temperature of 25 °C and 80% RH. 

To ensure proper identification of stage and host insect from which each parasitoid species was collected, the different lepidopteran larvae were separated according to species before rearing and spodopterans that pupated already in the field and found in the leaf funnels, leaf axis, ears, or closed tassels were recorded as unknown species, then placed individually in the same type of vials used for eggs, and identity confirmed at lepidopteran adult emergence. Similarly, egg batches were separated according to species based on the collection site and isolated per batch. The cocoon of parasitoids emerging from respective lepidopteran larvae or pupae in the laboratory were placed individually in vials (7 cm height and 2.5 cm diameter) till parasitoid adult emergence. Some adults from parasitized eggs, larvae, or pupa were labeled according to location, stage, and species of Lepidoptera from which they emerged and preserved in 96% ethanol for identification purposes. Specimens of adult parasitoids were sent to the Natural History Museum in London, United Kingdom where the identification was done by Dr. Andrew Polaszek for Scelionidae and Trichogrammatidae and Dr Gavin Broad for Braconidae and Ichneumonidae. The remaining live adults in the laboratory were used to establish their respective colonies for biological tests for their acceptability and suitability on *Spodoptera* species respective stages.

### 2.3. Laboratory Rearing of Spodoptera Species Colonies 

Approximately 200 larvae of various instars for each of the two spodopteran species *viz*. *S. frugiperda* and *S. eridania* were collected from maize and other crop fields such as tomatoes, Amaranthus, grasses, and weeds around the IITA Cameroon station (3.86353° N, 11.4630° E and 764 m a.s.l). Attempts were made to rear and maintain colonies of both species on maize and amaranthus leaves. The larvae were placed individually to avoid cannibalism into ventilated plastic jars (used for rearing field-collected larvae) in the laboratory and fed with leaves from respective host plants. The pupae of each spodopteran species were collected and placed in a moistened Petri dish (9-cm diameter) in oviposition cages of size 80 cm × 50 cm × 70 cm. Sterile cotton soaked in tap water was placed in a Petri dish of 5.5-cm diameter inside each oviposition cage while honey was provided as droplets at the upper side of the cage as a food source for the emerging adults. Each oviposition cage was provided with ten surrogate stems each prepared by wounding wax papers in a spiral form as an oviposition substrate. A photoperiod of 12 L: 12D at room temperature of 25 °C and 80% RH was maintained in the rearing and oviposition room. After approximately two to three days, black-head-stage egg batches were collected from the oviposition cages and placed in the same type of vials used for field-collected egg batches with a plastic aerated lid. Eggs were monitored daily for hatching; as soon as the first instars emerged, they were transferred using a camel-hair brush onto soft and fresh leaves of the host plant in the jars described previously for the rearing of larvae. Laboratory colony of each *Spodoptera* species was maintained for about five generations before these experiments; it was infused at least once every three months with wild individuals emerging from field samples. The diet was changed as need be by transferring the larvae onto new leaves in clean jars until pupation. Petri dishes, cages, and plastic jars were labeled according to species and dates.

### 2.4. Laboratory Rearing of Larval Parasitoids from the Field

All the cocoons emerging from respective spodopteran larvae or pupae originating from field collection were placed in ventilated Petri-dishes of 9-cm diameter. These parasitoids were monitored for adult emergence. Adult parasitoids obtained were isolated in a Plexiglas cage (15 cm × 15 cm × 15 cm) ventilated on two opposite sides with fine mesh and provided with water and honey. A light source was provided for mating for 24 h before starting the exposition of armyworm immature stages. To establish colonies of parasitoids, for each parasitoid species, an adult female was provided with about 15 larvae (5–9 days-old) of each armyworm, with leaves of the host plant as a diet for larvae; larvae were renewed daily until the death of the female parasitoid. After exposure, larvae were placed in the same type of plastic jars used for rearing larvae, with host-plant leaf diet and monitored for parasitoid cocoon emergence. Diet was changed every two days until pupation of un-parasitized larvae. In each Plexiglas cage, the adult parasitoids were fed with honey droplets provided on the upper surface of the cage, and cotton soaked with distilled water was provided. The procedure was continuous to increase the population of parasitoids under the same room condition as for armyworm. All Petri dishes, cages, and jars were labeled according to host insect, parasitoid species, and dates.

### 2.5. Laboratory Rearing of Egg Parasitoids from the Field

All parasitoids emerging from respective armyworm species egg-batches brought back from the field were fed in the same container in which they were incubated by providing thin droplets of honey at the upper surface of the type of vials used for field-collected egg batches. A 1-mm probe was used to make holes on the lid for aeration and a fine mesh was used to fasten the lid to prevent the escape of the tinny adults. Newly laid individual egg batches found on surrogate stems from oviposition cage were removed with part of the surrogate stem and introduced into each vial (7 cm height and 2.5 cm diameter) containing live adults of egg parasitoids. The batch exposed in the vial was allowed to be parasitized for 24 h. After the exposition, each egg batch was removed and transferred into a new vial. All the vials were maintained at room conditions similar to those for rearing armyworms and monitored till adult emergence. All vials were labeled with dates, host insect, and parasitoid species name. 

### 2.6. Biological Performance of Larval Parasitoids on Fall Armyworms 

Colonies of parasitoid species were successfully established only for *Charops* sp., *C. icipe*, and *C. luteum* hence only their performance as parasitoids was tested. The two sexes were often not obtained at the same time during studies on *S. eridania*, hence the biological parameters of *Charops* sp., *C. icipe*, and *C. luteum* were studied only on *S. frugiperda* the most important maize spodopteran pest. For host suitability test, second to third-instar (10 days old) host larvae were introduced with small pieces of young maize leaf tissue with leaf funnels (2.5 cm by 10 cm) into a 15 cm × 15 cm × 15 cm Perspex cage containing each larval parasitoid species. Leaf funnels were fixed vertically to the bottom of the containers with a small piece of masking tape. Host densities were 15 to 60 larvae per cage for two to five cohorts of mated naïve females of each larval parasitoid species and labeled according to parasitoid species and date of introduction. Five containers (replicates) were kept at larval rearing conditions for 24 h to be parasitized. The parasitoids were fed in a similar way as during colony rearing and a light source provided stimulate mating before exposure to larvae and during the experimental period. Larvae were then removed and reared in the same type of plastic jars used during rearing spodopteran larvae and fed with a natural diet until the emergence of parasitoids or pupation of the host. A new set of larvae and fresh diet was introduced every 24 h for parasitism repeatedly for five times. The pupa or cocoon and development time to adult emergence and adult longevity were recorded. The number of adult emerging from each replicate were noted and was used to compute the parasitism rates. The set-up was arranged in a completely randomized design and vial, cages, jars labeled according to host insect, parasitoid name, date of experiments, and by replicate. Fifty larvae were set aside for control mortality (10 per replicate in five replications) and not exposed to the parasitoid to assess the natural mortality of the hosts under the same rearing conditions. The larvae were reared individually till pupation and the number of dead larvae were counted and divided by the total number of larvae per replicate and multiplied by 100 to get percent natural mortality.

### 2.7. Biological Performance of the Egg Parasitoid on Fall and Southern Armyworms 

During this study, only the biological performance of *T. remus* whose colony was successfully established was evaluated. For egg parasitism, the procedure for studying its biological performance was like the procedure described for the colony production. The experiment was replicated five times with 50 to 100 egg batches exposed per replicate. Since un-parasitized eggs hatch (at black-head-stage) within three days, neonates hatching from them were removed to prevent cannibalism of parasitized eggs by neonates of the same batch and counted to determine egg parasitism. After removing the neonates, the remaining eggs and those of other batches were monitored daily until all eggs of a batch were completely dark (not black-head-stage) indicating that they were parasitized. The rate of egg parasitism was calculated by dividing the number of parasitized eggs by the total number of eggs for every batch multiplied by 100 and taking the average rate of egg parasitism of all batches. The number of parasitized eggs and parasitoids adults obtained per batch were used to estimate the emergence rate by dividing the number of adult parasitoids by the number of parasitized eggs multiplied by 100. Egg batches that did not have their eggs parasitized were noted to determine batch parasitism by dividing the total number of parasitized batches by the total number of batches collected multiplied by 100. Eggs within the same batch that hatched to neonates and added to the number of adult parasitoids from the same batch were used to determine egg viability rate by dividing this number by the total number of eggs of the batch multiplied by 100. The other egg batches were set aside for control mortality and not exposed to the parasitoid to assess the natural mortality of the hosts under the same rearing conditions. The number of unhatched eggs were counted and divided by the total number of eggs per batch and multiplied by 100 to get percent natural mortality. Parasitoid development time (from exposure to adult emergence) was evaluated by recording the date of exposure and date of adult emergence, while the date the adults died were recorded to estimate adult longevity. Based on the length of antennae (shorter for female), the sex ratio was estimated and recorded as the percentage of females in the total number (female and males) of adult parasitoids. The experimental design and set up and labeling were the same as that of larval parasitoids. 

For acceptability studies, 100 females were isolated from a cage after mating for 2 h and placed individually in the same type of vials used for exposure of egg batches to adult *T. remus*. To each adult, a newly laid egg batch was offered for oviposition and observed for 2 h. After 2 h the proportion of adult females that accepted to parasitized and attack the egg mass offered to them for oviposition was noted.

### 2.8. Data Analyses

Data collected for the field survey were summarized by descriptive statistics (counts and percentages). Percent parasitism was calculated by dividing the number of parasitoids that emerged by the total number of larvae collected for field samples or by dividing the number of parasitoids that emerged during laboratory exposure by the total number of larvae incubated after exposure to parasitoids and multiplied by 100 [32]. The relative abundance of parasitoids was calculated by dividing the number of each parasitoid species by the total number of parasitoids collected according to categories (egg or larval parasitoids) and multiplied by 100 and non-parametric test applied to compare the relative abundance among the species. Percent parasitism for each districts were subjected to a non-parametric analysis (because of variability in field parameters) to compare parasitism between seasons using Chi-square test. Where significant differences were found the Dunn test was used to separate the means.

For data on biological parameters of the larval parasitoids in the laboratory, the normality was tested and all except parasitism rate showed a normal distribution. For the normally distributed data, analysis of variance (ANOVA) between the performance of the various parasitoid species was performed including non-reproductive mortality and the Tukey test applied for post-hoc means separation. The t-test was performed between the non-reproductive mortality and natural mortality. Due to their non-normal distribution, the Kruskal Wallis non-parametric test was used for data on parasitism rates, and Dunn Test was conducted for mean separation after multiple comparisons. 

Similarly, the Kruskal Wallis non-parametric test was conducted for comparison between FAW and SAW for the performance of *T. remus* in egg count, egg viability, development time, parasitism rate, adult emergence rate, sex ratio, adult longevity, and significance of non-reproductive mortality. The non-reproductive egg mortality was assessed in the best-performing parasitoid, *T. remus*, using Abbot’s formula [33]. The significance of the non-reproductive mortality was also analyzed by comparing the natural egg mortalities recorded in control with mortalities recorded in presence of the egg parasitoid using Kruskal Wallis. All analyses were performed in R software version 4.0.2.

## 3. Results

### 3.1. Composition and Abundance of Armyworm Parasitoids

Six species of parasitoids were found from the field study; two egg parasitoids and four larval parasitoids (Figure 2). The egg parasitoids were *T. remus* and *Trichogramma chilonis* (Ishi) and the larval parasitoids were *Charops* sp., *C. luteum*, *C. sesamiae,* and *C. icipe*. All these species were identified to species level by NHM except for *C. sesamiae* which is already reported [34] and *Charops* which is likely a new species. The specimens from Cameroon were compared with all world species except *Charops flavipes* (Brullé) whose holotype could not be obtained by the museum during the time of this work. The Cameroonian species was different from all the world *Charops* species obtained. Between the two egg parasitoids, *T. remus* was the most abundant with a relative abundance of 93.1% (11,140 individuals; χ^2^_1,40)_ = 12.77.; *p* = 0.0004). It is extremely tinny (only 0.5 mm in length) and invisible to naked eyes in the field as well as in the laboratory, but is also the most widely distributed. The second egg parasitoid *T. chilonis* comes from a group of species that, however, are similar to each other and are also very small in body size (about 0.5 mm) with 101 individuals (7.7%). The most abundant larval parasitoid was *C. icipe* with 133 individuals (65%) followed by *Charops.* sp. with 48 (24.8%), C. sesamiae with 20 (7.3% and lastly *C. luteum* was 3 (2.9%) (χ^2^_3,80)_ = 24.82; *p* < 0.0001; Figure 2).

### 3.2. Geographic Distribution of Armyworm Parasitoids

The distribution and combination of two or more parasitoids per location are presented in Figure 3. During the exploratory survey in 2017, one egg parasitoid *T. remus* was collected from three ecozones, Zone III covering both the west and the north-west regions (Foumbot and Kumbo), Zone IV covering south-west and Littoral regions (Kumba), and Zone V covering center, south, and east regions (Mbalmayo) out of the five agro-ecological zones found in Cameroon, while two larval parasitoid *Charops* sp. and *C. sesamiae* were identified from Zone IV (Kumba). In the second sampling in 2019, the egg parasitoid *T. remus* and one larval parasitoid *C. icipe* were found in the two agro-ecological zones that were involved (Zone IV and V). *Charops* sp. and *C. luteum* were found only in Zone V. In the last survey in 2020, a second egg parasitoid *T. chilonis* found in Mbalmayo and *T. remus* (from Mbalmayo and Ntui), and two larval parasitoids *Charops* sp. (from Monatele and Mbalmayo) and *C. icipe* (from Bafia, Edea, and Mbalmayo) were recorded only from Zone V. One location (Ntui) situated in Zone V recorded four parasitoids species (*T. remus, C. icipe, Charops* sp., and *C. luteum*) and this is the only location from where *C. luteum* was collected.

### 3.3. Distribution of Armyworms and Host Plants between Geographic Regions and Sharing of Parasitoids Hosts 

Two species of *Spodoptera* were collected, namely, *S. frugiperda* and *S. eridania* (Table 2) with 5521 and 2638 larvae collected respectively. The FAW was found in all locations on maize (99.96%) but also on amaranthus (0.04%) in Pouma and Edea all in the Littoral region within Zone IV (humid forest with mono-modal rainfall). SAW was found mostly in Edea on amaranthus, groundnuts, and maize, in Pouma on maize and amaranthus all in the Littoral region. It was also found in Yaounde, Center region (humid forest with bi-modal rainfall) on tomato, amaranthus, and tropical chickweed (TCW) *Drymaria cordata*. SAW was the most distributed among the host plants (Table 2); however, 100% plant damage was observed in Edea and 50% in a tomato nursery when not treated in Yaoundé. 

*Telenomus remus* was found in three agro-ecological zones during all three years of the surveys and in all ten locations except Pouma and Monatele from FAW all on maize. From SAW, it was found only in Zone IV (Edea) with forest and mono-modal rainfall and only on Amaranthus. *Trichogramma chilonis* was recorded only during the last survey in 2020 from FAW on maize in Mbalmayo in Zone V, while *C. sesamiae* was only found during the first survey in 2017 from FAW on maize in Zone IV (Kumba); *C. icipe* was also collected on FAW from only two zones (IV and V with respectively) and seven locations except in Kumba (Zone IV), Kumbo and Foumbot (Zone III) on maize, but on SAW it was found only on amaranthus in Zone IV (Edea and Pouma) and V (Yaounde), and on plantain in Zone V (Yaounde). *Charops* sp. was recorded in two zones, Zone IV (Kumba) and Zone V in five locations (Bafia, Mbalmayo, Monatélé, Ntui, Yaounde) out of seven locations from FAW on maize. On SAW, it was collected from tropical chickweed at Yaoundé (Zone V). *C. luteum* was found only in one location (Ntui) in Zone V (Table 3). During this study a hyperparasitoid *Elasmus flaviceps* Ferrière (Hymenoptera: Eulophidae) was recovered from *C. icipe*’s cocoons on SAW infesting amaranthus in Yaoundé in Zone V.

### 3.4. Field Parasitism Rates of Egg Parasitoids

The highest parasitism rate was obtained from egg parasitism of FAW by *T. remus* ranging in all seven locations from 52.6% in Ntui to 100% in Mbalmayo and Kumba on maize with more than 90% parasitism in the other four districts where eggs were collected (Table 4). In Mbalmayo, *T. chilonis* was also collected with 100% parasitism rates. The single egg batch of SAW obtained in Edea from amaranthus was parasitized while out of 36 egg batches of SAW obtained in Yaoundé from Amaranthus, 32 (88.9%) were parasitized and egg parasitism was 98.3% (Table 4).

### 3.5. Field Parasitism Rates of Larval Parasitoids

Among the larval parasitoids, parasitism by *C. icipe* ranged from 0.1 to 4.1% on FAW on maize and from 0.3 to 40.0% on SAW on amaranthus with the highest values recorded in the same Zone IV from Pouma and Edea respectively (Table 5). Both pests had the same parasitism rate of 4.1% on maize. Parasitism by *Charops* sp. ranged from 0.3 to 3.5% with the highest recorded in Yaoundé (Nkolbisson) followed by Bafia with 3.4% from maize on FAW. The only location from which parasitism of SAW by *Charops* sp. occurred was Nkolbisson from Tropical chickweed (2.3%) and amaranthus (0.9%). In the sole location Ntui in Zone V where *C. luteum* was found, the parasitism rate was 0.3% on FAW on maize. For *C. sesamiae* 0.5% parasitism was recorded on FAW on maize, from Zone IV (Kumba) (Table 5). *Cotesia icipe* and *Charops* sp. were the larval parasitoid recorded from the two spodopteran pests, while *C. luteum* and *C. sesamiae* were recorded only on FAW.

### 3.6. Parasitism Rate by Season for Agro-Eclogical Zone III and IV

*Trichogramma chilonis* was collected only once hence the results are presented only for *T. remus*. In the mono-modal highland savannah (Zone III), egg parasitism has been reported only for the dry season for both districts (Foumbot and Kumbo) visited since egg of FAW were found only in the dry season (Table 6). No SAW eggs were collected in this zone. No larval parasitoid was collected in this zone III and no SAW larvae were collected as well (Table 6).

On the contrary, in Zone IV (mono-modal warm humid forest) although eggs were collected in both seasons in the district of Edea (Littoral region), parasitism was observed only in the rainy season (Table 6). However, in the dry season one egg batch of SAW was collected in Zone IV still from Edea with all eggs parasitize by *Telenomus remus* (Table 6). There was a significant variation in the number of larvae collected between the seasons with more individual recorded during the rainy season (X^2^ = 3.8; *p* = 0.04). However, parasitism was not statistically different between the seasons for *C. icipe* (χ^2^_1,2)_ = 0.05; *p* = 0.62) while *Charops* sp. and *C. sesamiae* were recorded only in the dry season (Table 6). For SAW, only C. icipe was collected and in the dry season with no significant effect of season on the number of SAW larva collected (Table 6). 

### 3.7. Parasitism Rate by Season of Sampling for Agroecological Zone V 

Zone V (bi-modal warm humid forest) has two dry and two rainy seasons but parasitism was observed in all season with rate in the rainy ranging from 0 to 84.7 ± 10.3% for egg parasitism and from 0 to 78.8 ± 9.3 for egg batch parasitism; while rates in the dry season from 0 to 37.5 ± 26.5% for egg batch and 0 to 49.7 ± 35.1% for egg parasitism (Table 7). For SAW, eggs were collected in the rainy season only in Mbalmayo and in the dry season only in Yaounde but only the those in Yaounde were parasitized at rates of 98.3% and 88.9% for egg and batch parasitism respectively (Table 7). No significant differences were found among seasons in egg parasitism (*p* ≥ 0.05). For larval parasitoids, parasitism ranged from 0 to 1.9 ± 0.5% in the rainy season and from 0 to 51.3 ± 48.7% in the dry season but with no significant difference among seasons (Table 7). For SAW, larval parasitism was obtained only for *Charops* sp. and *C. icipe* in dry season 2 and only for Charops sp. in rainy season 1 but no significant differences among seasons was obtained (Table 7).

### 3.8. Laboratory Performance of Larval Parasitoids: Host Acceptability and Suitability

In the laboratory, the three larval parasitoids *Charops* sp., *C. icipe,* and *C. luteum* tested successfully parasitized and developed on *S. frugiperda*. When FAW larvae were exposed to adult parasitoids, the non-reproductive mortality for *Charops* sp was 12.2 ± 1.5%, that of *C. icipe* was 6.4 ± 2.6% and *C. luteum* was 3.8 ± 2.3% (F_2,12_ = 3.94; *p* = 0.048; Table 8). The natural mortality of FAW larvae was 0% for all three larval parasitoids and non-reproductive mortality was significant only for *Charops* sp. (t = 7.98; *p* = 0.001), but not for *C. luteum* (t = 1.63; *p* = 0.179) and *C. icipe* (t = 2.45; *p* = 0.07). *C. icipe* showed the highest parasitism rate of 55.9 ± 12.6% statistically different from *Charops* sp. (χ^2^_2,12_ = 6.26; *p* = 0.044; Table 8). *C. icipe* also had significantly the shortest development time from egg to pupa (11.3 ± 1.6 days; F_2,12_ = 6.90; *p* = 0.010). There was no significant difference between parasitoid species in pupa to adult and egg to adult durations and the duration of the biological cycle (*p* > 0.05; Table 8). The adult longevity for *Charops* sp. was 13.0 ± 3.1 days; *C. icipe,* 16.1 ± 1.3 days and for *C. luteum*, 7.5 ± 0.9 days (F_2,12_ = 5.35; *p* = 0.029; Table 8).

### 3.9. Laboratory Performance of the Egg Parasitoid T. remus: Host Acceptability and Suitability

The initial host submission of egg batches to *T. remus* for acceptability resulted in 93.0 ± 1.74% and 91.6 ± 2.06% of egg batches of the FAW and SAW respectively, successfully attacked by the egg parasitoid, with no significant difference between hosts (χ^2^_1,45_ = 2.155; *p* = 0.142).

For suitability, the egg parasitoid *T. remus* successfully parasitized *S. eridania* and *S. frugiperda* (Table 9). The average number of eggs per batch were 154.6 ± 5.4 and 145.7 ± 9.2 eggs (χ^2^_1,439_ = 5.6; *p* = 0.018) for *S. frugiperda* and *S. eridania* respectively. Egg viabilities of 64.3 ± 2.0% and 50.6 ± 2.8% (χ^2^_1,439_ = 10.0; *p* = 0.002) were recorded for *S. frugiperda* and *S. eridania* respectively (Table 9). The natural mortality of FAW eggs was 8.70 ± 5.8 and statistically lower than the mortality in presence of the egg parasitoid, 23.1 ± 2.2 (χ^2^_1,439_ = 4.39; *p* = 0.036). The significant non-reproductive mortality obtained was 9.99% on this pest. For SAW, the natural mortality of eggs was 50.0 ± 11.0 and statistically similar to the mortality in presence of the egg parasitoid where 52.7 ± 3.7 was obtained (χ^2^ _1,110_ = 0.0003; *p* = 0.987). There were also significant differences in parasitism rates, higher on FAW (χ^2^_1,432_ = 24.0; *p* < 0.001), development time, longer on SAW (χ^2^_1,386_ = 88.6; *p* < 0.001), but had similar adult emergence rate (χ^2^_1,310_ = 1.1; *p* = 0.300). The sex ratio of emerged adults was higher on SAW (χ^2^_1,303_ = 73.5; *p* < 0.001) and they lived longer on FAW (χ^2^_1,240_ = 17.9; *p* < 0.001; Table 9).

## 4. Discussion

Similarly, to most countries across the African continent, Cameroon agricultural authorities approved emergency actions against FAW built around chemical pesticides without initially assessing the complex of Spodopteran pests in the country nor the complex of natural enemies associated with it. This is the first work in the country to document both the complex of armyworms and their natural enemies.

This study reveals that the composition of armyworms in Cameroon is dominated by the fall armyworm on maize and southern armyworm on amaranthus, both in terms of abundance and distribution. However, among the two *Spodoptera* species recorded, *S. eridania* had the highest host plant range. A report by Montezano et al. (2018) indicated that fall armyworm is a polyphagous pest with a host range of about 353 recorded plant species in 76 families, whereas *S. eridania* was reported on 202 host plants species from 58 families [2]. In this study, FAW was found only on maize and amaranthus in Cameroon, with maize being by far the main host plant, with 99.96% of its larvae collected on this plant alone. On the other hand, in Cameroon SAW was obtained from amaranthus, groundnuts, maize, tropical chickweed, and tomato in this study, with amaranthus and tomato being the main hosts having 75% and 12% of its larvae respectively. The latter species was reported for the first time in Cameroon on tomato in 2017 [6]. Two years later, during the current study, it was found causing significant damage on amaranthus with some farmers experiencing 100% plant damage in Edea and 50% of a tomato nursery damage when not treated with insecticides in Yaoundé. This is also the first report of *S. eridania* on amaranthus in Cameroon and second in Africa after Benin [6]. Its presence on tropical chickweed *D. cordata* and groundnuts is also the first report in Cameroon, Africa, and the world. While *S. exempta*, *S. littoralis,* and *S. triturata* had already been previously reported in Cameroon [3,35], they were not found in the present study suggesting that more intensive prospection is still needed. The presence of other Spodopteran species in the system opens the avenue of potential natural enemies dueling with them and that could be recruited for the natural control of the invasive Spodopteran species. More extensive and intensive prospection studies are however warranted in the country and region for more exhaustive cataloging on spodopteran species complex.

Six parasitoids species, composed of two egg parasitoids and four larval parasitoids, were found during this study. While conducting similar work in three different countries in East Africa, four species were found [36]. Except for the egg parasitoids *T. remus* and *T**. chilonis*, as well as the gregarious larval parasitoid *C. sesamiae,* they found the other three larval endoparasitoids recorded in the current study, in addition to *Palexorista zonata* (Curran) (Diptera: Tachinidae) and one egg parasitoid *C. curvimaculatus* which was not found in our study. However, in Niger alone, eight parasitoids were recorded including three egg parasitoids *Trichogrammatoidea* sp., *Trichogramma* sp., and *Telenomus* sp.; one egg-larval parasitoid *Chelonus* sp.; four larval parasitoids, *Cotesia* sp., *Charops* sp., and unidentified ichneumonid and tachinid [29]. The presence of *T. chilonis* in Cameroon is the second report in Africa on fall armyworm after Kenya [35]. However, 18 species of *Trichogramma* were already recorded long ago, with eight *Trichogrammatoidea* and 11 *Telenomus* species from eggs of borer pests [37,38]. The egg parasitoids *T. remus* is also the main egg parasitoids of FAW in North and South America [22,23,24]. The current study is therefore significant for the development of biological control of spodopterans. However, such efforts will be guided by ecological factors such as climate, host plant, and host insect because the present study showed differences in distribution and abundance of the various parasitoid species in the study locations, agro-ecological zones, host plants, and host insects. Similar studies are therefore required not only countrywide in Cameroon but through all major maize production areas in Africa where FAW is a potential or real threat to maize production.

The egg parasitoid *T. remus* was found in all three agro-ecological zones and from eight out of eleven locations while *C. icipe* occurred in seven locations covering two zones and *Charops* sp. from six locations covering the same two zones as *C. icipe*. *Coccygidium luteum* was only in one Zone and one location like *C.*
*sesamia* and *T. chilonis.* Within Niger, *T. remus* was reported from four different locations [29]. These parasitoids appear to be the most distributed parasitoid of armyworms in Africa as they were reported in five different countries [15,28]. The current study is the first report of the gregarious *C. sesamiae* on *S. frugiperda*. More extensive and intensive surveys are warranted to associate natural enemies’ diversity and abundance with agro-ecological zones and particularly the potential contribution of the humid forest in the current diversity reported versus savannah and arid agro-ecological zones that were not covered in the present study.

*Telenomus remus* and *C. icipe* shared the two pests on maize and amaranthus. Similarly, many studies have also reported that the egg parasitoid *T. remus* is a potential biological control agent for three spodopteran, *S. frugiperda, S. cosmioides* and *S. eridania* [39,40,41]. Apart from spodopterans, *T. remus* has recently been reported on the African stalk borer *Busseola fusca* (Fuller) (Lepidoptera: Noctuidae) on maize [42]. On the other hand, *C. icipe* was reported on *S. exigua* and *S. littoralis* on amaranthus in Kenya [33,43,44]. It also shared the same host insects *S. eridania and S. frugiperda* like the other larval parasitoid *Charops* sp. where they also shared the same host plant (maize and amaranthus). Such overlapping of host plants, as well as host insects between and among egg and larval parasitoids, could represent an asset for natural control of FAW in the system as we hypothesize that this could build on synergetic contributions of each parasitoid species in the reduction of the pest population.

Field parasitism by egg parasitoids in this study was considerably high (50 to 100%) and similar to those (20 to 70%) in Kenya [37] recently and much higher than those reported in Niger [29] with rates of 0.06 to 33.39%. This suggests that these egg parasitoids should be considered excellent candidates among the biological control agents of FAW for augmentative release [36], but more preferable, measures should be taken to obtain effective conservation biological control. For larval parasitoids, field parasitism by *C. icipe* was higher on SAW than FAW. *Charops* sp. recorded similar parasitism on FAW and SAW while *C. sesamiae* and *C. luteum* recorded very low parasitism rates. Contrary to egg parasitoids, these results are far below the recent levels reported from other African countries [28,37] apart from *C. icipe* in Ethiopia that had rates of 45.3% parasitism, close to 40% recorded from SAW on amaranthus in Edea. Lower rates than ours were recorded in Niger [29], while in the eastern region of Ghana higher rates than our case were recorded for *C. luteum* and *Chelonus bifoveolatus* Szépligeti (Hymenoptera: Braconidae) [45]. More robust studies are required to assess the potential effect of agro-ecological zones, seasons, and farmer practices on these differences found across countries. However, across all these countries, we hypothesize that if proper measures are taken to prevent harmful pesticide use and promoting environmentally friendly practices, parasitism rates in the systems invaded by FAW will considerably increase. Indeed, the parasitoids reported here are indigenous parasitoids probably thriving on local *Spodoptera* species before the invasion. New associations being observed may therefore be at their beginning stage and can improve, provided efforts are made to boost natural control of the invasive FAW and SAW in Africa.

Parasitism by *T. remus* ranged from 55.4 to 72.9% in the laboratory. Parasitism rates around this range have also been reported in previous studies [46,47,48] ranging from 58.0% to 80.7% on *S. frugiperda*. In Brazil, the parasitism of *T. remus* on *S. frugiperda* collected from maize and fed on an artificial diet for 36 generations under laboratory conditions varied from 80 to 100% [49]. The release of 5 to 8 thousand *T. remus* per hectare resulted in 90% parasitism in *S. frugiperda* eggs in Latin America [22,23] and Venezuela [24]. Such efforts in augmentative biological control should be promoted in Africa to boost natural control.

In the highland savannah with monomodal rainfall, and the warm humid coastal forest also with mono-modal rainfall all with one rainfed cropping season, the number of egg batches collected in the dry season were relatively small, the hence effect of season on parasitism is inconclusive in these zones. Sufficient egg batches were collected in the bimodal warm and humid forest with two rainfed cropping season and two dry seasons only for FAW for the four seasons. The rates of egg parasitism was above 75% in both seasons but relatively higher during the rainy season. Studies in Brazil in January 2020 reported a seasonal occurrence of parasitoids with no *Trichogrammatoidea* sp. emerging from parasitized eggs in dry season [48]. Similarly relatively higher rates of larval parasitism were obtained in the rainy season although the low number of samples for the dry season does not strengthen the observation that the differences were not significant. Trends with relatively higher larval parasitism rates in the rainy season than in the dry have been reported in Mozambique [50] especially for *Charops* sp and *C. luteum* on *S. frugiperda*. More effort should be made to determine the factors responsible for this relative difference between seasons as reports [50] showed higher mortality of *C. luteum* emerging from dry larvae collected in the dry season. Between the two rainy season of the current study, higher parasitism rate were observed for both *C. icipe* and *Charops* sp during the major rainy season from March to June suggesting a seasonal occurrence of the parasitoid [48]. Contrary to our finding, higher parasitism rate was found in the minor rainy season than the major one in Ghana [51]. This contrary result could be due to the fact that the parasitism in the two season were not evaluated from the same locations.

*Telenomus remus* further induced significant non-reproductive mortality in FAW where an additional 9.99% mortality was recorded in the pest in presence of *T. remus* compared to the natural mortality of the pest. Similarly, significant non-reproductive mortality rates were reported in egg parasitoids, with 17.8% for *Telenomus podisi* Ashmead (Hymenoptera: Platygastridae) on the stink bug *Halyomorpha halys* and 81% for *Trichogramma pretiosum* (Hymenoptera: Trichogrammatidae) on *Pieris rapae* (L.) (Lepidoptera: Pieridae) [52]. On *S. frugiperda*, similar rates were also reported from larval parasitoids *Cotesia flavipes* Cameron (Hymenoptera: Braconidae) with 10.0 to 11.5% on FAW pupae but higher on FAW larvae (23 to 36%) and *C. sesamiae* with 8 to 38% FAW larvae, 3.9 to 20.6% FAW pupae in Kenya [53]. This significant non-reproductive mortality may be associated with injuries caused by (1) multiple visits by parasitoids, implying repeated host stinging (destructive probing) with or without oviposition [54], (2) the introduction of toxic substances [55], or 3) venom arrestment factor injection; and feeding by the parasitoid larva before host-death (destructive host feeding) [56]. Further studies are warranted to elucidate the mechanism through which *T. remus* induces non-reproductive mortalities in FAW eggs. Moreover, the mechanisms involved in non-reproductive mortality are also influenced by temperatures, parasitoid age, host density, host age, parasitoid species, and parasitoid host [57]. In contrast, no significant non-reproductive mortality was recorded with SAW eggs, probably due to the instability of the colony of this pest in the laboratory. Indeed, while only 8.7% natural mortality was recorded in FAW in the laboratory conditions in the present study, high natural mortality of 50.0% was recorded in SAW under laboratory conditions. Non-reproductive host mortality is an important component of biological control. In Kenya for instance, new association between *C. flavipes* and *C. sesamiae*, two indigenous stemborer parasitoids known for ages in the system, and the invasive FAW did not result in successful offspring [53], however high non-reproductive host-mortalities were recorded. Parasitoids’ induced host egg abortion is rarely explicitly accounted for, leading to the underestimation of the ecosystem services provided by biological control agents [52]. With the recent invasion of FAW in Africa and Asia, we hypothesize indigenous parasitoids associating with this invasive pest will at the beginning have a high component of non-reproductive mortality while they evolve with the pest to improve parasitism efficiency. We recommended therefore that non-reproductive mortalities be recorded for all measure indigenous parasitoids across agro-ecological zones invaded by FAW.

In the laboratory, we found the highest parasitism rate by *C. icipe* with 55.9 ± 12.6%, close to the parasitism rate of 59.5% on *S. exigua* [44]. However, a higher rate of 85.59% was obtained on *S. littoralis* [33]. On fall armyworm other reports indicate up to 65.0% parasitism by *C. icipe* [58]. These rates demonstrate the high potential of this larval parasitoid which is gradually being reported also from various countries in Africa. This potential of these parasitoids was further increased by the non-reproductive mortality especially for Charops sp with a significant rate of 12.2 ± 1.5% not significantly higher than 6.4 ± 2.6% for *C. icipe*. On *S. frugiperda*, varying rates were reported from larval parasitoids *C. sesamiae* with 8 to 38% FAW larvae in Kenya [53]. Similar rate were obtained for *C. icipe* for single females [44] against *S. exigua*, but higher rate were obtained for cohorts of five female against *S. exigua* [44] and against *S. littoralis* [33]. This differences could be associated with differences in host pest, suggesting that the effect of host species and host stage should be investigated. The significance of mortality in the presence of parasitoid is evidence that it should also be considered in the evaluation of the performance of larval parasitoids in an integrated management approach for *S. frugiperda.* The development time of *C. icipe* from egg to pupa was 11.3 ± 1.6 days and close to the 14.6 ± 0.1 days and 13.3 ± 0.08 days reported on *S. littoralis* and *S. exigua* respectively in Kenya [33,44]. Several studies had reported similar egg to pupa durations such as *C. marginiventris* (12–13 days) [27], *C. plutellae* (12.6 days) [59] on the diamondback moth, *C. chilonis* (Munakata) (12.5 days) [60], *C. rubecula* (13.48 days) [61] and *C. ruficrus* (Haliday) (13.2 days) [19]. Having a parasitoid with shorter development is profitable in mass rearing and for the rapid buildup of populations in the field.

Second instar larvae were also used in the current study for *C. luteum* resulting in 30.3 ± 17% parasitism. Although studies on the suitability of this parasitoid are limited, higher rates up to 90% have been reported [62] with 1-day-old L1 (4-days-old neonates) when *S. frugiperda* was used. The difference in the result could be explained by differences in host larval stages used, as 2nd to 3rd instars (10-days-old) were used in the current study. More investigations are needed to identify the best host stage for this parasitoid. The development time of *C. luteum* was longest (27.7 ± 1.7 days); taking 18.9 days from egg to pupa stage and 9.6 days from pupa to adult emergence. A very short development time of 9 days from egg to pupa stage has been reported [62], but the adults emerged after 9.6 ± 0.8 days similar to what was observed during the present study. The difference in the egg to larval development time could be due to the use of different larval stages and diet and further studies are warranted to elucidate this.

The parasitism by *Charops* sp. of 8.7 ± 3.2% was the lowest compared to the other two larval parasitoids studied using exposed 2nd to 3rd instar *S. frugiperda* larvae. This rate was low compared to the 50% recorded for an unknown species of *Charops* found emerging on the African tussock moth *Orgyia mixta* Snell in Kenya. (Lepidoptera: Lymantriidae) from 3rd instars [50]. Further studies are warranted to confirm whether the Cameroonian species of *Charops* is a new species and how the host developmental stage could affect its performance under laboratory conditions. The development time of *Charops* sp. was 27.0 ± 3.5 day with the egg to pupal period of 14.0 ± 0.5 days and the pupa to adult emergence time of 12.6 ± 3.4 days, but adult longevity was 13.0 ± 3.1 days. The result obtained in the current study on *S. frugiperda* is similar to those reported for the unknown species of *Charops* with 34 days on *O. mixta* in Kenya [50].

## 5. Conclusions

The present result demonstrates the ecosystem service capabilities that unknown natural enemies could play in various systems. In the present case, we discovered for the first time in Cameroon various parasitoids associated with the new invasive spodopteran pests under various agro-ecological zones. *Telenomus remus* particularly should be considered an excellent candidate due to its high parasitism rates coupled with significant non-reproductive mortality in the laboratory and under field conditions, its wide geographic and multiple host range. The larval parasitoids *C. icipe*, *C. luteum,* and *Charops* sp. while having so far relatively low parasitism rates under field conditions also hold high potential to boost natural control on the invasive pests where farming systems are conducive for their establishment and growth. Further studies on preferred host stages parasitized, host range, as well as the effect of temperature, are warranted to unravel their full potential. The presence of these invisible friends against armyworms in general and fall armyworm in particular needs to be conserved. This requires a strategic change of policymakers from emergency actions based on the use of chemical pesticides to sustainable actions based on integrated pest management, with emphasis on promoting natural control, biopesticides, botanicals, and local innovations that are environmentally friendly.

## Figures and Tables

**Figure 1 insects-12-00509-f001:**
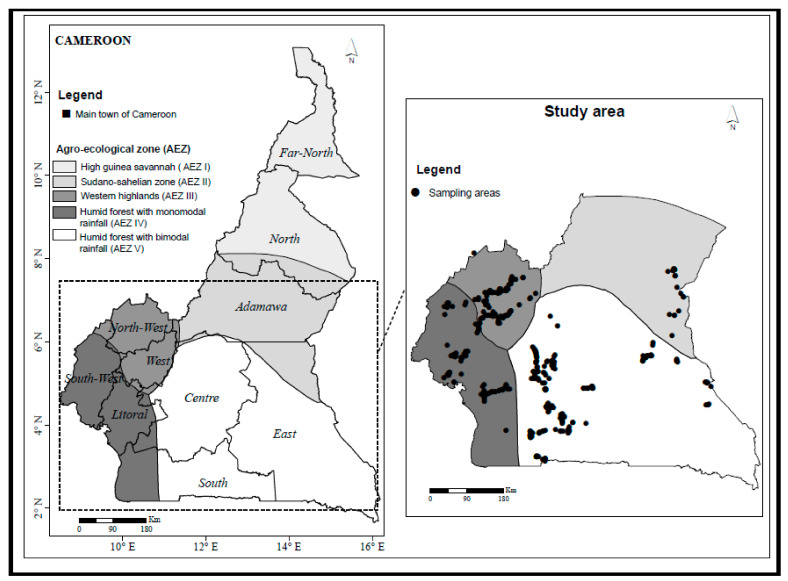
Map of Cameroon with names of its ten regional capitals (Extreme-north, North, Adamawa, North-West, South-West, West, Centre, Littoral, East, South), the five agro-ecological zones and the fields visited during the exploratory phase.

**Figure 2 insects-12-00509-f002:**
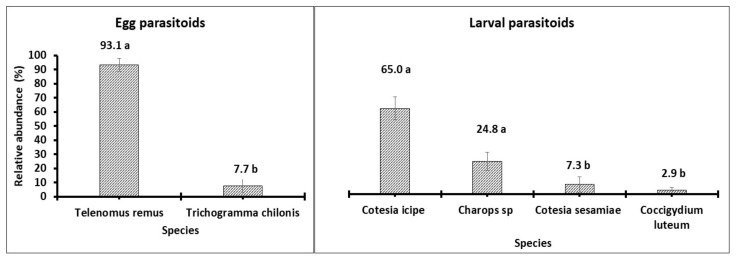
Relative abundance of parasitoid species collected from all the fields during the study.

**Figure 3 insects-12-00509-f003:**
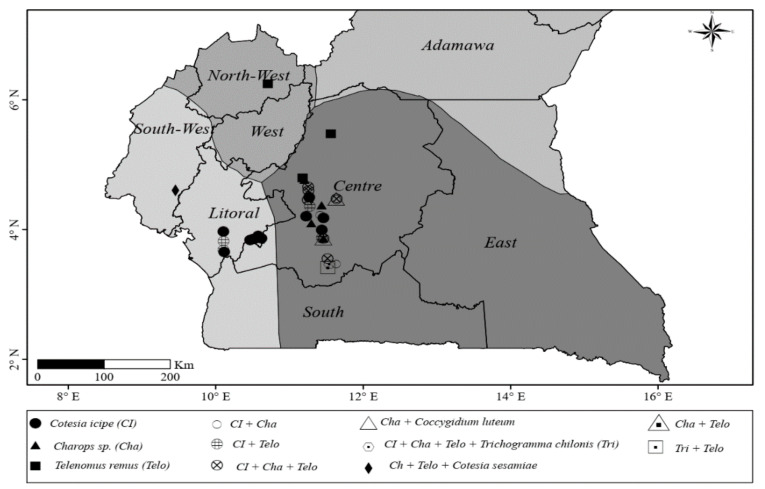
Distribution of armyworm parasitoids across years and among different agroecologies in Cameroon.

**Table 1 insects-12-00509-t001:** Number of fields sampled during the three-years surveys.

Region (Zone)	2017 Survey			2019 Survey			2020 Survey	
	No. 1	No. 2	No. 3	No. 4	No. 5	No. 6	No. 7	Total
Adamawa (Zone II)	4	5	2					11
West and Northwest (Zone III)	31	35	41					107
Littoral and Southwest (Zone IV)	19	20	26	17	7	12	7	108
Center, East and South (Zone V)	53	52	68	75	26	17	15	306
Total	107	125	188	92	33	29	22	596

Zone II: Guinea savannah (Adamawa region); Zone III: Western Highlands with savannah vegetation and mono-modal rainfall (West and North-West regions); Zone IV: warm and humid forest with mono-modal rainfall (Littoral and South-West regions); Zone V: warm and humid forest with bi-modal rainfall (Center, South, and East regions).

**Table 2 insects-12-00509-t002:** Number of spodopteran larvae and number of fields sampled for each host plant.

Host Plants	Number of Fields Sampled				Number of Larvae Collected			
	*S. eridania*	*S. frugiperda*	*S. litura*	Total	*S. eridania*	*S. frugiperda*	*S. litura*	Total
Amaranthus	10	5	0	15	1962	2	0	1964
Groundnuts	1	0	0	1	113	0	0	113
Maize	14	144	0	158	127	5519	0	5646
Plantain	0	0	16	16	0	0	168	168
TCW	6	0	0	6	130	0	0	130
Tomato	5	0	0	5	306	0	0	306
Grand Total	36	149	16	201	2638	5521	168	8327

TCW (Tropical chickweed).

**Table 3 insects-12-00509-t003:** Species of parasitoids and hyperparasitoid found on *Spodoptera* species and their geographic and agro-ecological distribution.

Species	Order: Family	Year	Ecozone	District	Host Plant	Host	Host Stage
*Telenomus remus*	Hym: Platygastridae	2017, 2019, 2020	III, IV, V	Ba, Ed, Fb, Ko, Mb, Nt, Ya	*A*spp, *Zm*	*Sf*, *Se*	E
*Trichogramma chilonis*	Hym: Trichogrammatidae	2020	V	Mb	*Zm*	*Sf*	E
*Coccygidium luteum*	Hym: Braconidae	2019	V	Nt	*Zm*	*Sf*	L
*Charops* sp.	Hym: Ichneumonidae	2017, 2019, 2020	V	Ba, Mb, Mo, Nt, Ya	*Zm*, *A*spp	*Sf*, *Se*	L
*Cotesia icipe*	Hym: Braconidae	2017, 2019, 2020	III, IV	Ba, Ed, Mb, Mo, Nt, Ya, Po	*Zm*, *A*spp, *Mp*	*Sf*, *Se*, *Ssp*	L
*Cotesia sesamiae*	Hym: Braconidae	2017	III	Ka	*Zm*	*Sf*	L
** Elasmus flaviceps*	Hym: Eulophidae	2020	V	Ya	*A*spp	*Ci*	P

Ba = Bafia, Ed = Edea, Fb = Foumbot, Ka = Kumba, Ko = Kumbo, Mb = Mbalmayo, Mo = Monatele, Ya = Yaounde, Nt = Ntui, Po = Pouma, *A*spp = *Amaranthus* spp., *Mp = Musa paradisiaca, Zm = Zea mays, Sf = Spodoptera frugiperda, Se = Spodoptera eridania, Ssp = Spodoptera sp, Ci = Cotesia icipe,* E = egg, L = larva, P = pupa, * Hyperparasitoid.

**Table 4 insects-12-00509-t004:** Rates of parasitism (%) by the two egg parasitoids on eggs and egg batches of the three spodopterans on various host plants, locations, and agro-ecological zones.

Ecozone	Location	Crop	Fields Sampled	Egg Batches Collected	*Telenomus remus* Parasitism (%)		*Trichogramma chilonis* Parasitism (%)	
					Batches	Eggs	Batches	Eggs
***Spodoptera frugiperda* (FAW)**								
Zone III	Foumbot	Maize	13	1	100	100	0	0
Zone IV	Kumbo	Maize	12	2	50	93.0	0	0
	Kumba	Maize	15	1	100	100	0	0
	Pouma	Maize	10	0	n/a	n/a	n/a	n/a
		Amaranthus	1	0	n/a	n/a	n/a	n/a
	Edea	Maize	17	11	81.8	90.8	0	0
		Amaranthus	1	0	n/a	n/a	n/a	n/a
Zone V	Bafia	Maize	17	17	88.24	93.22	0	0
	Mbalmayo	Maize	18	33	81.8	100	3.0	100
	Monatele	Maize	18	0	n/a	n/a	n/a	n/a
	Ntui	Maize	14	7	28.6	52.6	0	0
	Yaounde	Maize	10	12	83.3	97.4	0	0
***Spodoptera eridania* (SAW)**								
Zone IV	Pouma	Amaranthus	4	0	n/a	n/a	n/a	n/a
		Groundnuts	1	0	n/a	n/a	n/a	n/a
		Maize	10	0	n/a	n/a	n/a	n/a
	Edea	Amaranthus	2	1	100	100	0	0
		Maize	2	0	n/a	n/a	n/a	n/a
Zone V	Bafia	Maize	1	0	n/a	n/a	n/a	n/a
	Mbalmayo	Maize	1	3	0	0	0	0
	Yaounde	Amaranthus	4	36	88.9	98.3	0	0
		Tomato	5	0	n/a	n/a	n/a	n/a
		TCW *	6	1	0	0	0	0
***Spodoptera* sp.**								
Zone V	Yaounde	Musa spp	15	0	n/a	n/a	n/a	n/a
	Mbalamyo	Musa spp.	1	0	n/a	n/a	n/a	n/a

* TCW = Tropical chick weed (Drymaria cordata); n/a = not applicable; Zone II: Guinea savannah (Adamawa region); Zone III: Western Highlands with savannah vegetation and mono-modal rainfall (West and North-West regions); Zone IV: warm and humid forest with mono-modal rainfall (Littoral and South-West regions); Zone V: warm and humid forest with bi-modal rainfall (Center, South and East regions).

**Table 5 insects-12-00509-t005:** Rates of parasitism (%) by the four larval parasitoids on larvae of the three spodopteran pests on various host plants, locations, and agro-ecological zones.

Ecozone	Location	Crop	No. of Fields	No. of Larvae	*C. luteum*	*Charops sp.*	*C. icipe*	*C. sesamiae*
			Sampled	Collected				
***Spodoptera frugiperda***								
Zone III	Foumbot	Maize	13	277	0	0	0	0
	Kumbo	Maize	12	118	0	0	0	0
Zone IV	Kumba	Maize	15	217	0	0.5	0	0.5
	Pouma	Maize	10	123	0	0	4.1	0
	Edea	Maize	17	515	0	0	3.5	0
		Amaranthus		1	0	0	0	0
Zone V	Bafia	Maize	17	1189	0	1.6	1.2	0
		Amaranthus		1	0	0	0	0
	Mbalmayo	Maize	18	1008	0	0.5	1.4	0
	Monatele	Maize	18	881	0	0.3	2.6	0
	Ntui	Maize	14	1046	0.3	0.4	0.1	0
	Yaoundé	Maize	10	145	0	3.5	2.8	0
***Spodoptera eridania***								
Zone IV	Pouma	Amaranthus	4	1025	0	0	0.3	0
		Groundnuts	1	113	0	0	0	0
		Maize	10	123	0	0	4.1	0
	Edea	Amaranthus	2	15	0	0	40	0
		Maize	2	2	0	0	0	0
Zone V	Bafia	Maize	1	1	0	0	0	0
	Mbalmayo	Maize	1	1	0	0	0	0
	Yaoundé	Amaranthus	4	922	0	0.9	4.5	0
		Tomato	5	306	0	0	0	0
		TCW *	6	130	0	2.3	0	0
***Spodoptera*** **sp.**								
Zone V	Yaoundé	Musa spp	15	168	0	0	2.4	0
	Mbalmayo	Musa spp	1	0	n/a	n/a	n/a	n/a

* TCW = Tropical chick weed (Drymaria cordata); n/a = not applicable.

**Table 6 insects-12-00509-t006:** Rate of FAW and SAW parasitism (%) and number of egg batches or larvae collected in zone III and IV by season for each district.

Host Insect	Species	Zone III (Mono-Modal Highland Savannah Covering the North-West, and West Regions of Cameroon)				Zone IV (Mono-Modal Warm Humid Forest Covering the Littoral and Southwest Regions)			
		Dry	Rainy	X^2^	*p*-Value	Dry	Rainy	X^2^_1,3_	*p*-Value
FAW egg	Number of Egg batches	1.5 ± 0.5	0	2.67	0.10	1.0 ± 0.6	3.0 ± 3.0	0.05	0.82
	*T. remus* egg (%)	96.5 ± 3.5	-			0	90.8	2.00	0.16
	*T. rems* egg batch (%)	75.0 ± 25.0	-			0	100	2.0	0.16
FAW larva	Larvae collected	45.5 ± 20.5	122.0 ± 70.0	0.6	0.43	48.3 ± 26.3	136.7 ± 23.8	3.86	0.04
	*C. icipe* (%)	0	0			1.6 ± 1.6	1.9 ± 1.4	0.05	0.82
	*Charops* sp.(%)	0	0			0.3 ± 0.3	0	1.00	0.32
	*C. sesamiae* (%)	0	0			0.3 ± 0.3	0	1.00	0.32
SAW egg	Number of Egg batches					0.3 ± 0.3	0	1.00	0.32
	*T. remus* (%)					100	-		
	*T. chilonis* (%)					0	-		
SAW egg batch	*T. remus* (%)					100	-		
	*T. chilonis* (%)					0	-		
SAW larva	Larvae collected					367.3 ± 359.4	17.7 ± 17.2	0.19	0.66
	*C. icipe* (%)					18.9 ± 15.2	0	2.67	0.10
	*Charops* sp. (%)					0	0		
	*C. luteum* (%)					0	0		
	*C. sesamiae* (%)					0	0		

“-” (no collection); dry (dry season from November to February); rainy (Rainy season from March to October).

**Table 7 insects-12-00509-t007:** Rate of parasitism (%) and number of egg batches or larvae collected in zone V (Bi-modal warm humid forest covering the center, south, and east regions of Cameroon) by season.

Host Insect	Species	Dry 1	Dry 2	Rainy 1	Rainy 2	X^2^_3,6_	*p*-Value
Number of FAW Egg batches		1.3 ± 1.0	14.0 ± 14.0	11.2 ± 5.9	1.8 ± 1.1	3.15	0.37
FAW egg	*T. remus* (%)	49.7 ± 35.1	37.5 ± 37.5	84.7 ± 10.3	78.1 ± 9.8	1.81	0.61
	*T. chilonis* (%)	0	0	25.0 ± 22.4	0	1.50	0.68
FAW egg batch	*T. remus* (%)	37.5 ± 26.5	28.6 ± 28.6	78.8 ± 9.3	42.5 ± 11.1	3.90	0.27
	*T. chilonis* (%)	0	0	0.8 ± 0.7	0	1.50	0.68
Larvae collected		226.3 ± 77.7	38.5 ± 37.5	427.8 ± 116.5	244.8	4.60	0.20
FAW larva	*C. icipe* (%)	0.3 ± 0.2	51.3 ± 48.7	1.9 ± 0.5	1.1 ± 0.9	5.72	0.13
	*Charops* sp. (%)	0.7 ± 0.2	0	1.7 ± 0.9	0.5 ± 0.4	6.96	0.07
	*C. luteum* (%)	0	0	0.1 ± 0.1	0	2.0	0.57
	*C. sesamiae* (%)	0	0	0	0		
Number of SAW Egg batches		0	18.0 ± 0.0	1.0 ± 0.6	0	4.70	0.2.0
SAW egg	*T. remus* (%)	-	98.3	0	-	2.00	0.16
	*T. chilonis* (%)	-	0	0	-		
SAW egg batch	*T. remus* (%)	-	88.9	0	-	2.00	0.16
	*T. chilonis* (%)	-	0	0	-		
Larvae collected		4.0 ± 3.7	344.0 ± 344.0	71.2 ± 71.0	60.0 ± 60.0	1.12	0.77
SAW Larva	*C. icipe* (%)	0	6.0 ± 0.0	0	0	5.00	0.17
	*Charops* sp.(%)	0	1.2 ± 0.0	0.5 ± 0.3	0	3.75	0.29
	*C. luteum* (%)	0	0	0	0		
	*C. sesamiae* (%)	0	0	0	0		
Number of other spodopterans		2.8 ± 2.8	78.0 ± 78.0	0.2 ± 0.2	0	2.88	0.41
larvae	*C. icipe* (%)	0	2.6	0	-	2.0	0.37

“-” (no larvae collected); Rainy 1 (first rainy season from March to June); Dry 1 (First dry season from July to August); Rainy 2 (Second rainy season from September to October); Dry 2 (Second dry season from November to February).

**Table 8 insects-12-00509-t008:** Biological parameters of larval parasitoids in laboratory conditions (± SE).

Parameters	*Charops* sp.	*C. luteum*	*C. icipe*	X^2^ _2,12_	F_2,12_	*p*-Value
Parasitism rate (%)	08.7 ± 3.2 a	30.3 ± 17.0 ab	55.9 ± 12.6 b	6.26		0.044
Egg to Pupa (days)	14.0 ± 0.5 a	18.9 ± 1.9 b	11.3 ± 1.6 a		6.90	0.010
Pupa to adult (days)	12.6 ± 3.4	09.6 ± 0.8	11.0 ± 3.4		0.27	0.766
Egg to adult (days)	27.0 ± 3.5	27.7 ± 1.7	22.3 ± 4.9		0.67	0.530
Adult Longevity (days)	13.0 ± 3.1 ab	07.5 ± 0.9 a	16.1 ± 1.3 b		5.35	0.029
Biological cycle (days)	34.7 ± 2.2 b	34.7 ± 1.7 b	37.9 ± 5.8		0.18	0.841
Non-reproductive mortality	12.2 ± 1.5 b	03.8 ± 2.3 a	06.4 ± 2.6 ab		3.935	0.048

Mean followed by the same letter in same row are not significant different following Kruskal Wallis non-parametric test with Dunn post-hoc for parasitism rate and F-test with Tukey’s post-hoc for other parameters at *p* = 0.05.

**Table 9 insects-12-00509-t009:** Host acceptability and suitability of *Telenomus remus* on different *Spodoptera* species in laboratory.

Parameters	*FAW*	*SAW*	χ^2^	*df*	*p*-Value
Eggs per Batch	154.6 ± 5.4	145.7 ± 9.2	5.6	1, 439	0.018
Egg Viability (%)	64.3 ± 2.0	50.6 ± 2.8	10.0	1, 349	0.002
Development time (days)	12.1 ± 0.1	13.2 ± 0.1	88.6	1, 386	<0.001
Adult Emergence (%)	67.4 ± 2.1	73.7 ± 2.6	1.1	1, 310	0.300
Sex ratio	64.4 ± 1.3	88.5 ± 1.4	73.5	1, 303	<0.001
Longevity (days)	11.2 ± 0.3	8.5 ± 0.4	17.9	1, 240	<0.001
Parasitism rate (%)	72.9 ± 1.8	55.4 ± 2.6	24.0	1, 432	<0.001

Means separation by Dunn post-hoc following Kruskal Wallis non-parametric test with at *p* < 0.05.

## Data Availability

The data presented in this study are openly available at https://doi.org/10.25502/zb8y-bd92/d (accessed on 28 May 2021).

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
