# Peer review of "Natural Enemies of Fall Armyworm Spodoptera frugiperda (Lepidoptera: Noctuidae) in Different Agro-Ecologies"

_insects, 2021, doi:10.3390/insects12060509_

Round 1
Reviewer 1 Report
This is an important study and a very good manuscript. My only comment is that the scientists who identified the parasiotoids should be mentioned (not just the institution). They should also be thanked in the acknowledgements.
Reviewer 2 Report
The manuscript presents relevant conclusions and has scientific merit. The problem studied is of social importance and the theme is relatively new. I think that the results presented could serve as a basis for future more specific studies.
I will not present point-by-point considerations. However, I have a consideration that I think is the most relevant. The statistical analyzes and results (graphs and tables) seem to me to be slightly preliminary. As if it were descriptive results only (which does not detract from the work). For example, Figure 2 seems to me to be less rigorous from a scientific point of view. The tables, in general, seem to present raw data. Those collected from the field without any statistical treatment. What it seems is that the number of tables could be reduced and the information could be more condensed. The same could be said about the text of the results section. Percentages and rates are shown which are not accompanied by statistics. Fact that makes the result less conclusive and subjective.
Specifically in the discussion section (but also throughout the text of the manuscript), a lot of emphasis was placed on the regional problem which the manuscript seeks to solve. This is an important aspect. However, it gives the manuscript a regional character and may restrict readers' interest in general. My suggestion is to maintain the results, but in the discussion specifically, there is a tendency to take a more general approach to the results obtained. This will increase interest in the manuscript's conclusions.
What was said earlier, also applies to the title. Putting the local name of the work in the title can restrict the number of readers interested in the article. I believe that the name of the place of study could be removed from the title.
In general, I see that the manuscript is looking like a research report. It is necessary to transform it into a text of greater scientific interest. The results and conclusions are relevant and of broad interest. However, in the current format, the manuscript must undergo changes.
My final opinion is that the manuscript can be published as long as it meets the demands described above.
